# Relationship between Dietary Decanoic Acid and Coronary Artery Disease: A Population-Based Cross-Sectional Study

**DOI:** 10.3390/nu15204308

**Published:** 2023-10-10

**Authors:** Zhijian Wu, Weichang Yang, Meng Li, Fengyuan Li, Ren Gong, Yanqing Wu

**Affiliations:** 1Department of Cardiology, The Second Affiliated Hospital of Nanchang University, Nanchang 330006, China; wuzhijian3513@163.com (Z.W.); 363007220032@email.ncu.edu.cn (R.G.); 2Department of Respiratory Medicine, The Second Affiliated Hospital of Nanchang University, Nanchang 330006, China; yangweichang1996@163.com; 3Department of Respiratory Medicine, Nanchang First Hospital, Nanchang 330006, China; yeastud@163.com

**Keywords:** coronary artery disease, dietary decanoic acid, NHANES, cross-sectional study

## Abstract

Background: Coronary artery disease (CAD) is a cardiovascular disease with significant personal health and socioeconomic consequences. The biological functions of decanoic acid and the pathogenesis of CAD overlap considerably; however, studies exploring their relationship are limited. Methods: Data from 34,186 Americans from the National Health and Nutrition Examination Survey (NHANES) from 2003 to 2018 were analyzed. The relationship between dietary decanoic acid (DDA) and CAD prevalence was explored using weighted multivariate logistic regression models, generalized summation models, and fitted smoothing curves. Stratified analyses and interaction tests were conducted to explore the potential modifiers between them. Results: DDA was negatively associated with CAD prevalence, with each 1 g/d increase in the DDA being associated with a 21% reduction in CAD prevalence (odds ratio (OR) 0.79, 95% confidence interval (CI) 0.61–1.02). This relationship persisted after log10 and trinomial transformations, respectively. The OR after log10 transformation was 0.81 (95% CI 0.69–0.96), and the OR for tertile 3 compared with tertile 1 was 0.83 (95% CI 0.69–1.00). The subgroup analyses found this relationship to be significant among males and non-Hispanic white individuals, and there was a significant interaction (interaction *p*-values of 0.011 and 0.012, respectively). Conclusions: DDA was negatively associated with the prevalence of CAD, and both sex and race may modify this relationship.

## 1. Introduction

Coronary artery disease (CAD) is a prevalent cardiovascular condition and a significant contributor to disability and mortality worldwide, with significant personal health and socioeconomic impacts [1]. According to the World Health Organization, 17.7 million individuals died of cardiovascular disease (CVD) worldwide, accounting for 31% of the global mortality, with CAD accounting for approximately 7.4 million deaths [2]. Although studies have shown that risk factors such as age, hypertension, diabetes, dyslipidemia, smoking, and dietary habits contribute to the development of CAD [3,4], its exact etiology and pathological mechanism remain unclear. However, among the numerous risk factors, dietary habits and nutrient intake have received much attention in the development of CAD [5,6].

The ketogenic diet (KD) is a high-fat, low-carbohydrate diet that was first used in the treatment of epilepsy [7]. As research progressed, the KD has been found to be beneficial for a variety of diseases, such as Alzheimer’s disease, cancer, diabetes, and CAD [8,9,10,11]. However, the mechanism underlying the efficacy of KDs remains unclear. Moreover, several studies have demonstrated a poor association between serum ketone levels and disease control [12,13]. These results challenge the role of ketones themselves in disease control and raise the idea of a role for other components of the diet, particularly decanoic acid [14,15]. Decanoic acid, also known as caprylic acid, is a medium-chain saturated fatty acid commonly found in various dietary sources such as coconut oil, palm oil, dairy products, and fish (eel and salmon) [16,17]. Decanoic acid is also one of the major components of the KD and plays an important role in it. This may be because it is easily converted into ketones in the liver, to serve as an alternative energy source for the brain and other tissues [9]. In addition, decanoic acid has unique properties compared to other fatty acids. Studies have found that decanoic acid has powerful anticonvulsant properties, which are especially beneficial for people with epilepsy [14,18]. In addition, it is believed to have improved mitochondrial function, antioxidant, and regulated blood lipid and glucose levels [9,15]. Given the trend toward younger age and the enormous public health burden of CAD, it is urgent that effective interventions are available for prevention. Combined with the beneficial effects of a KD and the ability of decanoic acid to influence a variety of CAD risk factors, we hypothesized that dietary decanoic acid (DDA) could reduce the incidence of CAD. If our hypothesis proves to be correct, it will provide a new dietary strategy for the prevention and treatment of CAD and thus have a significant impact on public health.

Here, we extracted data on DDA and CAD prevalence from the representative National Health and Nutrition Examination Survey (NHANES) of 2003–2018, to investigate the correlation between DDA and CAD among American adults.

## 2. Materials and Methods

### 2.1. Study Design and Population

The National Health and Nutrition Examination Survey (NHANES) is a cross-sectional survey with a complex multistage sampling design that is conducted every 2 years. It is designed to provide information on the health and nutritional status of a representative U.S. population. The NHANES research plan received ethical approval from the National Center for Health Statistics ethics review board. Prior to participating, all individuals provided written informed consent. To find out more, please visit the website: www.cdc.gov/nchs/nhanes/index.htm (accessed on 28 August 2023).

This study used NHANES data from eight survey cycles from 2003 to 2018 to explore the association of DDA with CAD in U.S. adults. A total of 80,312 participants were recruited in these cycles. The following participants were excluded: those <18 years of age (*N* = 32,549), who were pregnant (*N* = 868), missing or had incomplete data on DDA (*N* = 9905), and missing data on CAD history (*N* = 2804). Finally, 34,186 participants were included in the analysis (Figure 1).

### 2.2. Dietary Decanoic Acid

The NHANES nutritional assessment component was performed by highly trained dietary surveyors who were proficient in both English and Spanish. The participants were asked about all the food and beverages they had consumed in the last 24 h and specific food scales. Pictures were used to help the participants to estimate the amount of food they had consumed. The obtained data were then used to calculate each participant’s energy, nutrient, and other food component intake through specialized software. The first dietary recall interview took place face-to-face in a mobile examination center (MEC), and the second was conducted through a telephone interview 3–10 days later. In this study, the total estimate of DDA (grams/day, g/d) was calculated as the mean of two recall periods.

### 2.3. Coronary Artery Disease

As in previous NHANES studies, the interviews in this NHANES questionnaire were performed by well-trained health professionals, and every issue was standardized. Participants were either interviewed at home or at an MEC and were asked the following questions: “Have you ever been told by a doctor or other health professional that you have CAD?”, “Have you been told by a doctor or other health professional that you have angina pectoris?”, and “Have you been told by a doctor or other health professional that you have a myocardial infarction?”. If they answered “yes” to one or more of these questions, they were considered to have CAD [19,20].

### 2.4. Potential Covariates

In this study, covariates that could change the estimated effect of the DDA by >10%, or those that were considered traditional risk factors for CAD, were collected as potential confounding variables for subsequent statistical analyses [21]. The following variables were included: dietary nutrition including energy (kcal), protein (g/d), carbohydrate (g/d), fiber (g/d), total fat (g/d), saturated fatty acids (SFA, g/d), monounsaturated fatty acids (MUFA, g/d), polyunsaturated fatty acids (PUFA, g/d), and multivitamins; demographic characteristics including sex, age, race, education, and poverty income ratio (PIR); physical measurements including body mass index (BMI), waist circumference, and blood pressure; laboratory tests including alanine aminotransferase (ALT, U/L), serum creatinine (SCR, umol/L), blood urea nitrogen (BUN, mg/dL), estimated glomerular filtration rate (eGFR, mL/min/1.73 m^2^), uric acid (UA, umol/L), fasting blood glucose (FBG, mmol/L), glycosylated hemoglobin (HbA1c, %), total cholesterol (TC, mmol/L), and triglyceride (mmol/L); and questionnaire surveys including smoking status, alcohol use status, hypertension, and diabetes. Among them, the eGFR was calculated based on the modification of diet in renal disease formula [22]. Hypertension was defined as a self-reported physician diagnosis of hypertension, systolic blood pressure ≥ 140 mmHg and/or diastolic blood pressure ≥ 90 mmHg, or the use of antihypertensive medication [23]. Diabetes was defined as a self-reported physician diagnosis of diabetes, FBG ≥ 7 mmol/L, or HbA1c > 6.5% [24].

### 2.5. Statistical Analysis

To compare the baseline characteristics of the with and without CAD groups, a weighted Student’s *t*-test or Mann–Whitney test was used for continuous variables, and a chi-square test for categorical variables. The continuous variables are summarized as mean ± standard deviations, and the categorical variables are summarized as counts (percentages, %). Because the DDA is skewed, for testing the robustness of the results, the statistical analyses in this study simultaneously presented the raw DDA data, log10-transformed data, and tertiles of the data. A weighted multivariate regression analysis was conducted to investigate the association between CAD prevalence and DDA, and three models based on DDA were conducted in this study: Model 1 was unadjusted; Model 2 was adjusted for sex, age, race, education levels, and PIR; and for Model 3, BMI, hypertension, diabetes, smoking status, drinking status, TC, triglycerides, UA, SCR, eGFR, BUN, ALT, energy, protein, carbohydrate, fiber, total fat, SFA, MUFA, PUFA, and multivitamins were taken into account, in addition to the adjusted factors in Model 2. Shapes describing the association of DDA (raw and log10 transformed data) with CAD prevalence were generated using a generalized additive model (GAM) and smoothed curve fitting (penalized spline method). Furthermore, stratified analyses and interaction tests were conducted for the following variables to assess the potential variables modifying the relationship between DDA and CAD prevalence: sex (male or female), age (<60 or ≥60 years), race (Mexican American, other Hispanic, non-Hispanic white, non-Hispanic black, or other race), educational levels (<9th grade, 9–11th grade, high school, college, or graduate and above), BMI (<25 kg/m^2^, 25~29.9 kg/m^2^, or >30 kg/m^2^), hypertension (no or yes), diabetes (no or yes), smoking status (has never smoked, quit smoking or current smoker), drinking status (has never drunk, 1~5 drinks/month, 5~10 drinks/month, >10 drinks/month, and unknown), and eGFR (<60 or ≥60 mL/min/1.73 m^2^).

For the missing categorical variables, we added dummy variables (16.72% (*N* = 5717) were missing for drinking status), and for missing continuity variables, we interpolated them using multiple interpolations. Overall, all data, except for drinking status, were only marginally missing (<10%). The distribution of variables with missing data compared to the raw data is shown in Appendix A. Statistical analyses were conducted using the R Pack (http://www.R-project.org (accessed on 28 August 2023)) and EmpowerStats (http://www.empowerstats.com (accessed on 28 August 2023), X&Y Solutions, Inc., Boston, MA, USA). A two-sided *p* < 0.05 was considered statistically significant.

## 3. Results

### 3.1. Baseline Characteristics of the Study Population

Overall, 34,186 individuals were recruited in this study, with a mean age of 50.47 years. Of these, 7.85% (*N* = 2685) suffered from CAD, and the median DDA was 0.356 (interquartile range 0.191–0.584). The participants’ baseline characteristics, weighted to categorize them into groups with and without CAD, are presented in Table 1. Except for ALT, Vitamin A, and Vitamin C, all other baseline characteristics were significantly different between both groups. In the CAD group, the mean age was 65.61 years, and 59.91% were male. Similar to previous studies, the patients in the CAD group were more likely to be older, male, non-Hispanic white, smokers, and non-drinkers, have hypertension and diabetes, have higher levels of BMI, HbA1c, FBG, BUN, SCR, UA, and triglycerides, and have lower levels of PIR, education, eGFR, TC, and DDA, and lower nutrient intake including total energy, protein, fiber, fatty acids and multivitamins compared with the non-CAD group.

### 3.2. Association between DDA and CAD Prevalence

Table 2 summarizes the results of the weighted multiple regression analysis. Overall, the relationship between the DDA and CAD prevalence was negative. In the unadjusted model, the relative odds of CAD prevalence decreased by 40% with each 1 g/d increase in the DDA (odds ratio (OR) 0.60, 95% confidence interval (CI) 0.49–0.72). This negative correlation in Model 2 (OR 0.69, 95% CI 0.56–0.84) and Model 3 (OR 0.89, 95% CI 0.61–1.02) remained robust after adjusting for the confounding factors. This relationship remained stable when the continuous DDA was log10-transformed as well as converted into tertiles of the DDA to minimize the effect of skewness and outliers on the results. In the fully adjusted model, the prevalence of CAD decreased by 19% (OR 0.81, 95% CI 0.69–0.96) with each natural logarithmic increase of 1 g/d DDA and decreased by 15% (OR 0.85, 95% CI 0.75–0.97) in tertile 2 (0.24 ≤ DDA < 0.49, g/d) and 17% (OR 0.83, 95% CI 0.69–1.00) in tertile 3 (0.49 ≤ DDA < 5.03, g/d) compared to those in tertile 1 (0.00 ≤ DDA < 0.24, g/d). Consistent results were obtained for the data collected using the multiple interpolation method. The ORs of the pooled estimates for these three types of data in the fully adjusted model were 0.758 (95% CI 0.631–0.910), 0.800 (95% CI 0.763–0.946), and 0.860 (95% CI 0.707–0.906), respectively (Appendix A). Consistently, the GAM and smoothed curve fitting showed a negative relationship between DDA and CAD prevalence (Figure 2A,B).

### 3.3. Subgroup Analyses

In the subgroup analyses, the association between the DDA and the prevalence of CAD remained significantly negative in the following subgroups: males, non-Hispanic white individuals, those with the educational attainment of graduation or higher, individuals with a BMI of less than 25 kg/m^2^, individuals without hypertension, diabetes, quit smokers, individuals who consumed five or more alcoholic beverages per month, and those with a high eGFR (estimated glomerular filtration rate) of 60 or greater. However, there was a significant interaction only for males and non-Hispanic white individuals (*p*-values for the interaction were 0.011 and 0.012, respectively) (Figure 3).

## 4. Discussion

In this study, the results of the weighted multivariate regression analyses and smoothed curve fitting showed that the DDA (either raw, log-transformed, or tertile data) was significantly and negatively associated with the prevalence of CAD. Further analysis, found that both sex and race, as variables, would likely modify the above relationship.

Over the past few decades, with economic and social developments, people have become increasingly conscious of their health status, bringing the KD to the forefront of attention. While the KD was initially developed to control epilepsy, it is now widely used for a variety of diseases such as Alzheimer’s, cancer, CVD, diabetes, and obesity [8,9,25,26,27]. Two common types of KDs are the classic KD and the medium-chain triglyceride (MCT) KD [9]. Therefore, exploring the mechanisms behind the efficacy of the KD is important. According to previous research, the primary therapeutic mechanism of the KD is to replace carbohydrates as a main source of energy by producing large amounts of ketones [28]. On the contrary, some studies have shown that although KD is effective in disease treatment, the serum ketone levels correlate poorly with disease control [12,13,29]. With the development of the MCT KD, attention turned to medium-chain fatty acids, especially decanoic acid [15,30]. It has been suggested that decanoic acid improves the function and number of mitochondria in neuronal cell culture systems and leads to the transcriptional up-regulation of genes related to lipid metabolism and down-regulation of genes related to glucose metabolism [15,31]. In another animal study, decanoic acid-rich mustard oil was found to improve blood lipids, enhance antioxidant protection, and reduce lipid peroxidation compared to natural mustard oil [32].

Although there is significant overlap between these biological polymorphisms of decanoic acid and the pathogenesis of CAD, to the best of our knowledge, few studies have explored the relationship between decanoic acid and CAD to date. This study evaluated the relationship between DDA and the prevalence of CAD among American adults using data from a total of 34,186 participants over eight NHANES cycles. The findings suggested that a higher DDA intake was associated with lower rates of CAD in the population, in line with previously published research on the beneficial effects of KD and decanoic acid in humans [9,14,15,25]. This may be attributed to the following mechanisms. First, decanoic acid increases the production of ketones (beta-hydroxybutyric acid, acetone, and acetoacetate). Ketones have various anti-inflammatory properties, including the inhibition of the NLRP3 inflammasome [33,34], reduction in the levels of cytosolic pro-inflammatory factors (IL-1β, IL-6, IFN-γ, MCP-1) [35], and inhibition of oxidative stress [36]. Inflammation is known to be integral to the development of CAD, initiating the early stages of atherosclerosis and continuing throughout the process. Second is the direct action of caprylic acid. Recent studies have demonstrated that caprylic acid has several biological activities in vivo, including the inhibition of diacylglycerol kinase (DGKA) [37] and the activation of peroxisome proliferator-activated receptor gamma (PPAR-γ) [15]. DGKA is a key enzyme involved in the final step of the triacylglycerol synthesis, combining fatty acids with glycerol to form triacylglycerols, which in turn adversely affects the development of CAD [38]. PPAR-γ is a transcription factor that inhibits the inflammatory response, promotes lipid metabolism, and has anti-atherosclerotic effects, thereby exerting a protective effect against CAD [39]. In addition, decanoic acid is known to inhibit phosphoinositide signaling and α-amino-3-hydroxy-5-methyl-4-isoxazolepropionic acid receptor receptors [14,40]. However, it remains unclear whether this effect has an impact on CAD, and therefore, further research should be conducted in the future.

Following the recommendations of the STROBE statement, we conducted further subgroup analyses to represent the dataset more appropriately. Our findings show that DDA remained significantly negatively associated with CAD prevalence in both male and non-Hispanic white populations, and there were interactions. This sex difference may be explained by differences in individual sex hormone levels. It was shown that men have more androgens in their bodies, which may increase their risk of developing CAD [41], and in women, estrogen in the body provides some cardiovascular protection before menopause [42]. In addition, men are more likely to have unhealthy lifestyle habits such as smoking, alcohol consumption, obesity, and physical inactivity, all of which may increase the risk of CAD. Therefore, men are more likely to benefit from DDA. Variations in genetic risk factors, obesity status, drinking, and other factors may provide potential interpretations of significant race-specific differences. Further large-sample prospective studies are needed to explore the relationship between DDA and CAD in non-Hispanic black individuals. Interaction tests show that race variables may modify this relationship. No other variables were found to significantly modify this relationship. This implied that the negative correlation between DDA and CAD is consistent regardless of age, sex, weight, blood pressure, or other relevant factors, which could contribute to the wider application of DDA as an important nutrient for the prevention of CAD.

Although this was a large-sample study that adequately accounted for the complex stratified sampling design of NHANES, it has some inherent limitations. The first is its cross-sectional study design. The design was not as comprehensive as a cohort study, and although we attempted to minimize the effects of the confounders, we could not completely exclude the effects of other confounders on the results. Moreover, causal inferences could not be made, and the ability to explore etiology was limited. Second, we used self-reported dietary intake data, which may be subject to memory bias and measurement errors, and the same is true for the definition of CAD. However, it is worth noting that questionnaires are an important part of national health and nutrition surveys, and numerous studies have been conducted based on questionnaire data. Third, we were unable to exclude participants who were breastfeeding due to the lack of breastfeeding data in our database. This may have had an impact on our findings because breastfeeding individuals may have specific physiological or behavioral characteristics. Therefore, future studies need to take this into account to improve the accuracy and comprehensiveness of the study. Fourth, this study was limited to adults in the United States, and extrapolation to other countries should be carried out with caution. Finally, it is important to note that because this study did not have data on whether participants were on a KD or not, it is not possible to arrive at a conclusion about the effects of a KD on CAD. In addition, it is unclear whether a KD is needed to increase the DDA to levels that are beneficial for CAD. In summary, more prospective studies are needed to validate these results before translating them into clinical applications.

## 5. Conclusions

DDA was negatively associated with the prevalence of CAD, implying that DDA is protective against CAD. Moreover, subgroup analyses showed that DDA remained significantly negatively associated with CAD prevalence in both males and non-Hispanic populations, and there were interactions. However, because of the inherent limitations of cross-sectional studies in general and those of the present study, future well-designed prospective and experimental studies to validate the results of this study are warranted.

## Figures and Tables

**Figure 1 nutrients-15-04308-f001:**
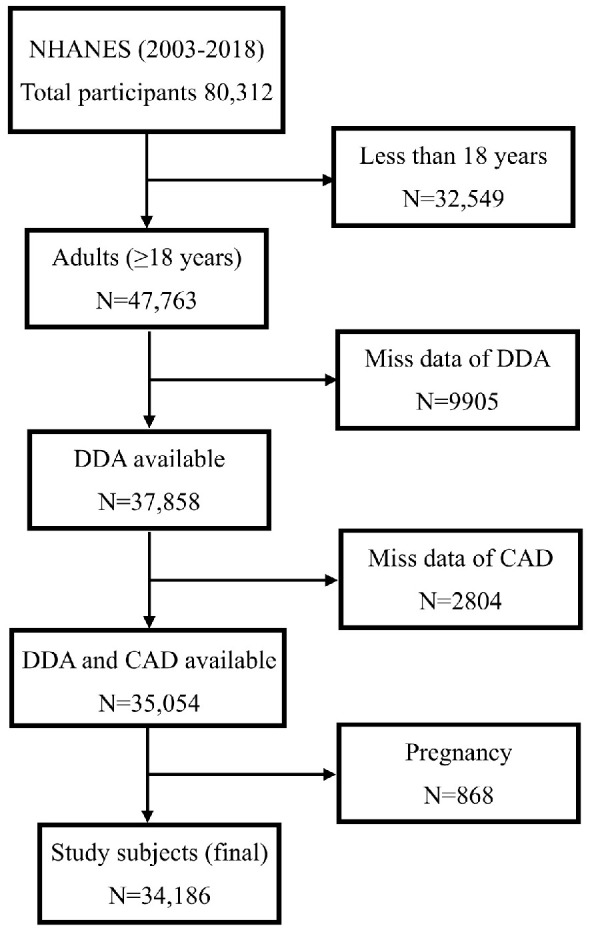
Flow chart of participants. Abbreviations: DDA dietary decanoic acid, CAD Coronary artery disease.

**Figure 2 nutrients-15-04308-f002:**
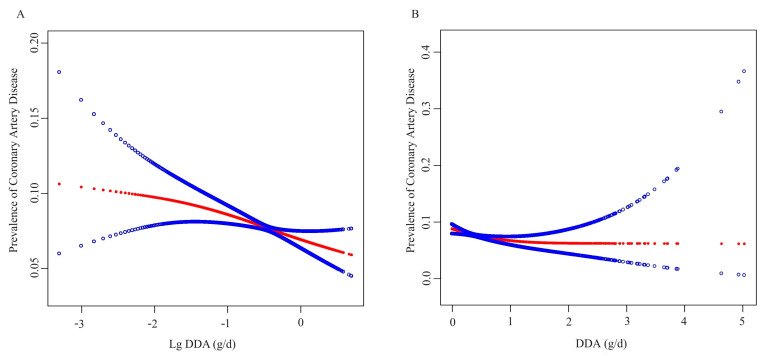
(**A**) The association of DDA (log 10 converted) with the prevalence of CAD, (**B**) The association of DDA (raw data) with the prevalence of CAD. The red line and blue line represent the estimated values and their corresponding 95% confidence intervals, respectively. Sex, age, race, education levels, PIR, BMI, hypertension, diabetes, smoking status, drinking status, TC, triglyceride, BUN, UA, SCR, eGFR, ALT, energy, protein, carbohydrate, fiber, total fat, SFA, MUFA, PUFA, and multivitamins were adjusted.

**Figure 3 nutrients-15-04308-f003:**
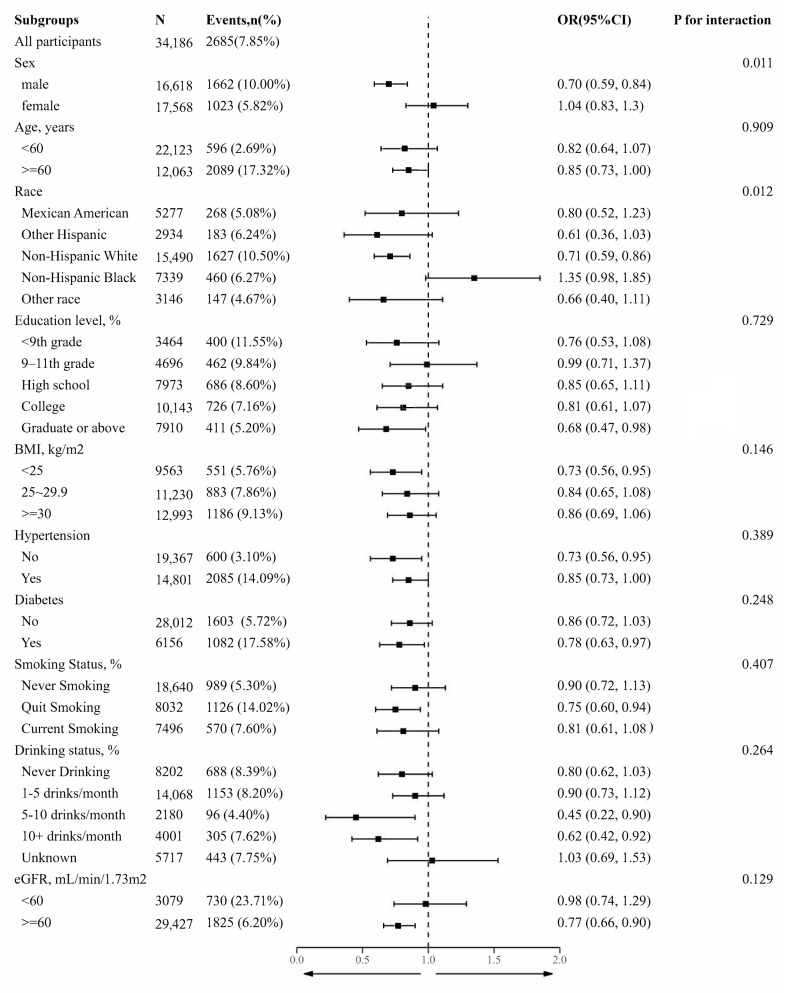
Stratified analyses according to potential modifiers of the association between DDA and the prevalence of CAD. Each subgroup analysis was adjusted for sex, age, race, education levels, PIR, BMI, waist circumference, hypertension, diabetes, smoking status, drinking status, FBG, TC, triglyceride, BUN, UA, SCR, eGFR, HbA1c, ALT, energy, protein, carbohydrate, fiber, total fat, SFA, MUFA, PUFA, and multivitamins, except for the stratifying variable. * Numbers that do not add up to 100% are attributable to missing data.

**Table 1 nutrients-15-04308-t001:** Weighted characteristics of the study population.

Variables ^a,^*	Non-CAD (*N* = 31,501)	CAD (*N* = 2685)	*p*-Value
Age, years	46.85 ± 16.43	65.61 ± 12.49	<0.0001
Sex, %			<0.0001
Male	47.15	59.91	
Female	52.85	40.09	
Race, %			<0.0001
Mexican American	8.05	3.70	
Other Hispanic	5.17	3.15	
Non-Hispanic white	69.05	78.90	
Non-Hispanic black	11.07	8.57	
Other race	6.67	5.69	
Education level, %			<0.0001
<9th grade	4.80	8.66	
9–11th grade	9.87	14.24	
High school	23.42	28.02	
College	31.66	29.56	
Graduate or above	30.26	19.53	
PIR	3.07 ± 1.63	2.70 ± 1.56	<0.0001
BMI, kg/m^2^	28.93 ± 6.81	30.37 ± 6.81	<0.0001
Hypertension, %			<0.0001
No	64.51	25.07	
Yes	35.43	74.93	
Diabetes, %			<0.0001
No	88.02	62.77	
Yes	11.98	37.23	
Smoking status, %			<0.0001
Never smoking	55.86	35.65	
Quit smoking	22.16	41.94	
Current smoking	21.98	22.41	
Drinking status, %			<0.0001
Has never drunk	19.65	23.61	
1–5 drinks/month	41.06	42.33	
5–10 drinks/month	7.86	3.42	
10+ drinks/month	14.13	12.90	
Unknown	17.30	17.73	
FBG, mmol/L	5.44 ± 1.74	6.31 ± 2.38	<0.0001
HbA1c, %	5.57 ± 0.87	6.11 ± 1.14	<0.0001
TC, mmol/L	5.10 ± 1.07	4.64 ± 1.19	<0.0001
Triglyceride, mmol/L	1.70 ± 1.49	1.91 ± 1.29	<0.0001
ALT, U/L	25.37± 18.54	24.78 ± 33.37	0.1943
BUN, mg/dL	4.77 ± 1.83	6.13 ± 2.95	<0.0001
UA, umol/L	320.66 ± 82.13	353.00 ± 91.18	<0.0001
SCR, umol/L	78.17 ± 30.13	94.16 ± 47.65	<0.0001
eGFR, mL/min/1.73 m^2^	92.84 ± 23.57	76.25 ± 24.51	<0.0001
Energy, kcal	2113.30 ± 829.80	1904.67 ± 758.04	<0.0001
Protein, g/d	82.71 ± 36.51	74.36 ± 32.95	<0.0001
Carbohydrate, g/d	252.80 ± 111.84	228.45 ± 98.73	<0.0001
Fiber, g/d	16.80 ± 9.10	15.96 ± 8.66	<0.0001
Total fat, g/d	82.04 ± 40.15	74.22 ± 37.10	<0.0001
SFA, g/d	26.68 ± 14.61	24.04 ± 12.88	<0.0001
MUFA, g/d	29.22 ± 13.87	26.63 ± 14.09	<0.0001
PUFA, g/d	18.36 ± 9.85	16.89 ± 9.96	<0.0001
Vitamin E, mg/d	8.37 ± 5.69	7.57 ± 4.94	<0.0001
Vitamin A, μg/d	647.79 ± 566.36	623.16 ± 489.08	0.0658
Vitamin B6, mg/d	2.11 ± 1.41	1.90 ± 1.35	<0.0001
Vitamin C, mg/d	82.86 ± 79.01	80.80 ± 78.80	0.2369
Vitamin K, μg/d	114.01 ± 215.49	99.73 ± 107.70	0.0029
DDA, g/d	0.49 ± 0.37	0.43 ± 0.35	<0.0001

Abbreviations: PIR, poverty income ratio, BMI, body mass index, FPG, fasting plasma glucose, HbA1c, hemoglobin A1c, BUN, blood urea nitrogen, TC, total cholesterol, UA, uric acid, SCR serum creatinine, eGFR, estimated glomerular filtration rate, ALT, alanine aminotransferase, SFA, saturated fatty acids, MUFA, monounsaturated fatty acids, PUFA, polyunsaturated fatty acids, DDA, dietary decanoic acid. ^a^ Missing treatment: categorical variables (adding dummy variables), continuous variables (multiple interpolations). * Data are presented as mean ± standard deviation and numbers (%) as appropriate.

**Table 2 nutrients-15-04308-t002:** Weighted relative odds of CAD according to DDA in different models among American adults.

DDA, g/d	Coronary Artery Disease OR (95% CI), *p*-Value		
	Model 1	Model 2	Model 3
DDA	0.60 (0.49, 0.72) < 0.001	0.69 (0.56, 0.84) < 0.001	0.79 (0.61, 1.02) 0.078
Lg DDA	0.66 (0.59, 0.74) < 0.001	0.74 (0.64, 0.84) < 0.001	0.81 (0.69, 0.96) 0.015
DDA Tertile			
T1 (0.00–0.24)	Reference	Reference	Reference
T2 (0.24–0.49)	0.79 (0.71, 0.88) < 0.001	0.82 (0.73, 0.93) 0.002	0.85 (0.75, 0.97) 0.014
T3 (0.49–5.03)	0.66 (0.58, 0.75) < 0.001	0.75 (0.65,0.86) < 0.001	0.83 (0.69, 1.00) 0.056
P for trend	<0.001	<0.001	0.047

Abbreviations: DDA, dietary decanoic acid. Lg DDA value was log10-transformed. Model 1 adjusts for no variables; Model 2 adjusts for sex, age, race, PIR, and education levels; Model 3 adjusts for sex, age, race, education levels, PIR, BMI, hypertension, diabetes, smoking status, drinking status, TC, triglyceride, BUN, UA, SCR, eGFR, ALT, energy, protein, carbohydrate, fiber, total fat, SFA, MUFA, PUFA, and multivitamins.

## Data Availability

A publicly available dataset was analyzed for this study, which can be accessed by visiting https://wwwn.cdc.gov/nchs/nhanes/Default.aspx (accessed on 28 August 2023).

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
