# Peer review of "Relationship between Dietary Decanoic Acid and Coronary Artery Disease: A Population-Based Cross-Sectional Study"

_nutrients, 2023, doi:10.3390/nu15204308_

Round 1
Reviewer 1 Report
Overall
· The sentence structures/grammar could be improved.
Abstract
· Line 19-21: Please improve this sentence “There was a linear negative correlation between DDA and the prevalence of CAD for each unit increase in DDA, the prevalence of CAD decreased by 24% (OR 0.76, 95% CI 0.62 ~ 0.92). “
· Line 22-23: In “the ORs were 0.76 (95% CI 0.67 ~ 0.87) and 0.76 (95% CI 22 0.66 ~ 0.89), respectively.” please mention the category for each OR.
· Line 22-23: Please change these findings “In subgroup analyses, no variable was found to significantly modify this relationship (all interaction P > 0.05)”. We can see significant association in some subgroups in Figure 3. You need to report the results of subgroup analysis in parentheses. Interaction tests and subgroup analysis are separate tests and are needed to report their results separately.
Introduction:
· Do you want to assess decanoic acid as part of ketogenic diet or regular diet? If you want to assess in regular diet, please mention the dietary sources of decanoic acid in the introduction.
· Line 43: Reference formatting, not in []
· Line 51: Decanoic acid is a saturated fatty acid
· Line 51: Please correct this sentence “Decanoic acid is a polyunsaturated fatty acid” since Decanoic acid is a ten-carbon, saturated fatty acid.
· Line 56: For this study, it seems that prevalence of CAD has been assessed. Please correct this sentence “For this study, we extracted data on dietary decanoic acid (DDA) and CAD incidence from the representative National Health and Nutrition Examination Survey (NHANES) from 2003 through 2018.”.
· Elaborate more on relationship between decanoic acid and the ketogenic diet and why you’ve selected decanoic acid specifically
Methods:
· Exclusion criteria: Did you exclude participants reported currently being pregnant, breast feeding, and the participants without complete data on two 24-h dietary recalls?
· Consider adding energy intake, physical activity, aspirin use, multivitamin use and energy intake as covariates. multivitamin use among covariates? They are important factors when CAD is assessed.
· Line 64-66: Sentence is awkwardly written, suggest rephrasing
· 111-113: Sentence says “physician’s self-reported data” this should be reworded. It is self-reported data from participants based on if they been told by a medical provider that they have hypertension
· Line 119-120: “Summarized continuous variables in mean ± 119 standard deviation (SD) and categorical variables in counts (percentages, %).”: This sentence seems incomplete.
· BMI and waist circumference fall under the category of anthropometric measurements, while diabetes and HbA1c are indicators of blood sugar levels. It is more effective to target one measurement from each group, such as diabetes and BMI, for adjustment and monitoring.
Results
· Similar to methods, some important covariates are missing in the adjusted models including physical activity, aspirin use, multivitamin use, and energy intake
· In the assessment of the association between diet and CAD, it is crucial to consider important confounding factors such as physical activity, energy intake, supplement use, aspirin use, and dietary components like dietary fiber, fruits, and vegetable intake. It is advisable to include these variables in Table 1 and account for their effects by adjusting for them in Table 2.
· Line 166-167: When reporting the linear association between DDA and CAD, it is advisable to specify the DDA dosage. Please add the dose in this sentence “In the unadjusted model, the CAD prevalence showed a decreasing trend with increasing DDA (167 0.63, 95% CI 0.53 ~ 0.76).”
· Please provide a detailed account of the subgroup analysis results. Specifically, identify the subgroups in which significant associations were observed.
· Please provide nutrient intake including the energy percent from dietary fat, protein, carbohydrate, PUFA, MUFA, and SFA in table 1.
Discussion
· Have you assessed serum ketone bodies in this study? Evaluating serum ketone bodies is valuable because they represent one potential mechanism underlying the association between DDA and CVD.
· Is DDA taken into consideration in the context of a high-fat diet or a regular diet?
· Please mention the results of subgroup analysis in the discussion.
· Line 214: Does this mean generalizability to the general population?
· What are the study strengths?
· English language revision is needed as there are some portions that are unclear and various incomplete sentences.
Author Response
- Overall: The sentence structures/grammar could be improved.
Reply: Thank you for carefully reviewing this manuscript and for your valuable comments, and we apologize for any grammatical/structural errors that may exist. We tried our best to improve the manuscript and made some changes to the manuscript. These changes will not influence the content and framework of the paper. Here we did not list the changes but marked them in red in the revised paper. A certificate of editing in scientific English is attached. We appreciate for Editors/Reviewers’ warm work earnestly and hope that the correction will meet with approval.
- Abstract:
2.1 Line 19-21: Please improve this sentence “There was a linear negative correlation between DDA and the prevalence of CAD for each unit increase in DDA, the prevalence of CAD decreased by 24% (OR 0.76, 95% CI 0.62 ~ 0.92).
Reply: We have revised the sentence in the article. However, because of the addition of covariates, the ORs change but the direction remains the same. The detail is as follows: DDA was negatively associated with CAD prevalence, with each 1 g/d increase in DDA being associated with a 15% reduction in CAD prevalence (OR 0.85, 95% CI 0.69, 1.06). (Line 18-20)
2.2 Line 22-23: In “the ORs were 0.76 (95% CI 0.67 ~ 0.87) and 0.76 (95% CI 22 0.66 ~ 0.89), respectively.” please mention the category for each OR.
Reply: We have changed it to “The OR after log10 transformation was 0.84 (95% CI 0.73 ~ 0.97), and the OR for tertile 3 compared with tertile 1 was 0.85 (95% CI 0.72 ~ 1.00)” in the revised manuscript. Thank you for your attention to detail and for making the statement clearer. (Line 21-23)
2.3 Line 22-23: Please change these findings “In subgroup analyses, no variable was found to significantly modify this relationship (all interaction P > 0.05)”. We can see significant association in some subgroups in Figure 3. You need to report the results of subgroup analysis in parentheses. Interaction tests and subgroup analysis are separate tests and are needed to report their results separately.
Reply: We are grateful for the suggestion. As suggested by the reviewer, we have changed it to “In stratified analyses, DDA was negatively associated with CAD prevalence in all but the Non-Hispanic Black population. Interaction tests show that race variables may modify this relationship (P for interaction = 0.023)” (Line 23-25)
- Introduction:
3.1 Do you want to assess decanoic acid as part of ketogenic diet or regular diet? If you want to assess in regular diet, please mention the dietary sources of decanoic acid in the introduction.
Reply: Thank you greatly for your helpful suggestions for the quality improvement of this manuscript. In this study, we aim to assess the impact of decanoic acid consumption on the prevalence of CAD in individuals following a regular diet. To more clearly reflect the purpose of this study, we have added a more detailed interpretation regarding the source of the decanoic acid in the revised manuscript: Decanoic acid, also known as caprylic acid, is a medium-chain saturated fatty acid commonly found in various dietary sources such as coconut oil, palm oil, dairy products, and fish (eel, and salmon). (Line49-51)
3.2 Line 43: Reference formatting, not in []
Reply: We have revised.
3.3 Line 51: Decanoic acid is a saturated fatty acid
Reply: We thank the reviewer for pointing this out. We have fixed the error.
3.4 Line 51: Please correct this sentence “Decanoic acid is a polyunsaturated fatty acid” since Decanoic acid is a ten-carbon, saturated fatty acid.
Reply: We apologize for our mistake. We have changed it to “Decanoic acid, also known as caprylic acid, is a medium-chain saturated fatty acid”.(Line 49-50)
3.5 Line 56: For this study, it seems that the prevalence of CAD has been assessed. Please correct this sentence “For this study, we extracted data on dietary decanoic acid (DDA) and CAD incidence from the representative National Health and Nutrition Examination Survey (NHANES) from 2003 through 2018.”
Reply: We thank the reviewer for pointing this out. We have changed incidence to “prevalence”.
3.6 Elaborate more on relationship between decanoic acid and the ketogenic diet and why you’ve selected decanoic acid specifically
Reply: We are grateful for this suggestion. In order to provide more clarity on the reviewers' concerns, we have added more information on the relationship between caprylic acid and the ketogenic diet and why caprylic acid was chosen over other fatty acids (because decanoic acid has a variety of biological functions which are strongly associated with the development of CAD). The sentences are as follows: Decanoic acid is also one of the major components of the ketogenic diet and plays an important role in it. This is because it can be easily converted into ketones in the liver to serve as an alternative energy source for the brain and other tissues. Secondly, decanoic acid has unique properties compared to other fatty acids. Studies have found that decanoic acid has powerful anticonvulsant properties, which are especially beneficial for people with epilepsy. In addition, decanoic acid is thought to have improved mitochondrial function, antioxidant, and regulating blood lipid and glucose levels. Although these effects are also important in inducing coronary atherosclerosis, the specific effects of decanoic acid on CAD in individuals following a regular diet have not been thoroughly investigated. Therefore, this study aims to fill this research gap and provide insights into the potential role of decanoic acid in CAD. (Line 51-62)
- Methods:
4.1 Exclusion criteria: Did you exclude participants reported currently being pregnant, breastfeeding, and the participants without complete data on two 24-h dietary recalls?
Reply: Thanks for the heads up. Indeed, the nutritional intake and needs of those in a pregnant state and breastfeeding are different from those of the general population. Excluding these populations can be helpful in minimizing bias in the results, however, unfortunately, the NHANES database only collects pregnancy status and no data on breastfeeding. Therefore in the next analysis, we excluded those who were pregnant and the final study results were unchanged.
To ensure a large sample size and to minimize selection bias, we excluded only participants with missing data from both dietary recall interviews and statistically processed this data as follows: the total estimate of DDA (g/d) was calculated as the mean of two recall periods (if only the first interview data were collected, it was used instead of the mean value). A similar method is used for other dietary data.(Line 95-96)
4.2 Consider adding energy intake, physical activity, aspirin use, multivitamin use, and energy intake as covariates. multivitamin use among covariates? They are important factors when CAD is assessed.
Reply: Thank you very much for your interest in this issue and for providing some important factors. As the reviewer notes, energy intake, physical activity, aspirin use, and multivitamin use are indeed important considerations when assessing the relationship between dietary nutrition and coronary heart disease. We reincorporated energy, protein, fatty acids, fiber, and multivitamins as covariates. Although the multivariate regression results are similar to the previous ones, increasing more convincing.
But regarding physical activity, aspirin use was not present in every cycle of NHANES data, so we did not include these covariates. We also believe that this does not affect the results of the study. This is because the study also suggests that not all covariates associated with the dependent variable should be included in the multiple regression model. The selection of which covariates to include in the model is based on the purpose of the study, theoretical basis, data availability, and statistical analysis considerations.
4.3 Line 64-66: Sentence is awkwardly written, suggest rephrasing
Reply: We sincerely thank the reviewer for careful reading. We have changed it to “The National Health and Nutrition Examination Survey (NHANES) is a cross-sectional survey with a complex multistage sampling design that is conducted every two years. It is designed to provide information on the health and nutritional status of a representative U.S. population”. (Line 71-74)
4.4 111-113: Sentence says “physician’s self-reported data” this should be reworded. It is self-reported data from participants based on if they been told by a medical provider that they have hypertension
Reply: We feel sorry for our carelessness. In our resubmitted manuscript, we have changed it to “Hypertension was defined as a self-reported physician diagnosis of hypertension”. Thanks for your correction.(Line 121-122)
4.5 Line 119-120: “Summarized continuous variables in mean ± 119 standard deviation (SD) and categorical variables in counts (percentages, %).”: This sentence seems incomplete.
Reply: Thanks for your careful checks. As suggested by the reviewer, we have corrected it into “Continuous variables are summarized as mean ± standard deviation (SD) and categorical variables are summarized as counts (percentage, %)”(Line 128-129)
4.6 BMI and waist circumference fall under the category of anthropometric measurements, while diabetes and HbA1c are indicators of blood sugar levels. It is more effective to target one measurement from each group, such as diabetes and BMI, for adjustment and monitoring.
Reply: We think this is an excellent suggestion. We were identifying covariates based on traditional risk factors for CAD and previous studies. There is no doubt that diabetes, BMI, waist circumference, or HBA1c are risk factors for CAD. Therefore, these factors were included in the multivariate regression analyses, but this raises some problems such as covariance and over-adjustment (although no significant covariance was found during covariate screening). Therefore, we included only BMI and diabetes in the adjusted model based on the reviewers' comments.
- Results:
5.1 Similar to methods, some important covariates are missing in the adjusted models including physical activity, aspirin use, multivitamin use, and energy intake
Reply: Thank you very much for your valuable advice. As the reviewer notes, energy intake, physical activity, aspirin use, and multivitamin use are indeed important considerations when assessing the relationship between dietary nutrition and coronary heart disease. We reincorporated energy, protein, fatty acids, fiber, and multivitamins as covariates. Although the multivariate regression results are similar to the previous ones, increasing more convincing.
Because physical activity, aspirin use was not present in every cycle of NHANES data, we did not include these covariates. We also believe that this does not affect the results of the study. This is because the study also suggests that not all covariates associated with the dependent variable should be included in the multiple regression model. The selection of which covariates to include in the model is based on the purpose of the study, theoretical basis, data availability, and statistical analysis considerations.
5.2 In the assessment of the association between diet and CAD, it is crucial to consider important confounding factors such as physical activity, energy intake, supplement use, aspirin use, and dietary components like dietary fiber, fruits, and vegetable intake. It is advisable to include these variables in Table 1 and account for their effects by adjusting for them in Table 2.
Reply: Thanks again for providing these important covariates and factors. We included these factors in Table 1 to compare the differences between the two groups. And these factors were included in Table 2 as covariates to be adjusted in the multivariable regression analysis to more accurately assess the relationship between DDA and CAD and to exclude the interference of other factors.
5.3 Line 166-167: When reporting the linear association between DDA and CAD, it is advisable to specify the DDA dosage. Please add the dose in this sentence “In the unadjusted model, the CAD prevalence showed a decreasing trend with increasing DDA (167 0.63, 95% CI 0.53 ~ 0.76).”
Reply: Thanks to your comment, we rewrote the sentence as follows: In the unadjusted model, the relative odds of CAD prevalence decreased by 37% with each 1 g/d increase in DDA (OR 0.63, 95% CI 0.53 ~ 0.76).(Line 181-183)
5.4 Please provide a detailed account of the subgroup analysis results. Specifically, identify the subgroups in which significant associations were observed.
Reply: Thank you very much for your suggestion, we have written the results of the subgroup analysis in detail in the results section as follows: In subgroup analyses, except for non-Hispanic Black participants, DDA (log 10 transformed) was negatively associated with CAD prevalence in all of the following subgroups: sex, age, race, edu-cation levels, BMI, hypertension, diabetes, smoking status, drinking status, and eGFR (≥60 mL/min/1.73 m2). Similarly, among these variables, only race significantly modified the relationship between DDA and CAD prevalence (P-values for interaction = 0.023) (Line 212-217)
5.5 Please provide nutrient intake including the energy percent from dietary fat, protein, carbohydrate, PUFA, MUFA, and SFA in Table 1.
Reply: Thank you very much for your suggestion, we have put all these factors in Table1. Overall, nutrient intake (including total energy, protein, fiber, fatty acids, and multivitamins) was lower in the CAD group compared to the non-CAD group.
- Discussion
6.1 Have you assessed serum ketone bodies in this study? Evaluating serum ketone bodies is valuable because they represent one potential mechanism underlying the association between DDA and CVD.
Reply: We appreciate the reviewer’s insightful suggestion and agree that it would be useful to assess serum ketone bodies in this study. Unfortunately, due to resource constraints, the NHANES database does not have data about serum ketones. Although serum ketones may have an important potential mechanism between DDA and CAD, it does not affect our findings. Nevertheless, given the potential role of ketone levels between DDA and CAD, we will explore this aspect in depth in the future (animal and human levels). Thank you again for your valuable suggestions to improve the quality of our manuscript.
6.2 Is DDA taken into consideration in the context of a high-fat diet or a regular diet?
Reply: Thank you for your question. Our study was conducted in a representative general population, so this study will consider DDA for regular dietary. In addition, in accordance with the reviewer's comments, more information about decanoic acid including its source, biological function, and potential association with CAD has been included in the introduction section of this manuscript.
6.3 Please mention the results of subgroup analysis in the discussion.
Reply: Thank you very much for your helpful suggestions! We have added more information to the discussion section as follows: Following the recommendations of the STROBE statement, we conducted further subgroup analyses to represent the data set more appropriately. Our findings show that DDA is negatively associated with CAD prevalence in almost all subgroups except Non-Hispanic Black participants. Interaction tests show that race variables may modify this relationship. Variations in genetic risk factors, obesity status, drinking, and other factors may provide potential interpretations of significant race-specific differences. Further large-sample prospective studies are needed to explore the relationship between DDA and CAD in Non-Hispanic Black individuals. Interaction tests show that race variables may modify this relationship. No other variables were found to significantly modify this relationship, implying that the negative correlation between DDA and CAD is consistent regardless of age, sex, weight, blood pressure, or other relevant factors, which could contribute to the wider application of DDA as an important nutrient for the prevention of CAD. (Line 272-284)
6.4 Line 214: Does this mean generalizability to the general population?
Reply: Thank you for your question. This study invoked weighting variables in the regression model to minimize sample heterogeneity, reduce bias, and optimize sample representation. However, the study sample was limited to the American adult population. In conclusion, the results of this study can be applied to the entire American adult population, but extrapolation to other countries requires caution.
6.5 What are the study strengths?
Reply: First, this study utilized data from a large, representative population to investigate the relationship between DDA and CAD prevalence for the first time, filling a gap in this research area. Second, we demonstrated the reliability of the findings by invoking weighting variables in the regression analysis, as well as by applying multiple statistical treatments to the DDA (log10 and tertile transformation). Third, we found a protective effect of DDA on the prevalence of CAD, which, if demonstrated by well-designed prospective trials in the future, could guide the implementation of new nutritional interventions for the prevention of CAD. At last, the results of the study can provide a basis for the development of functional foods and drugs associated with DDA, which is critical to public health.

Reviewer 2 Report
Dear Authors:
I read with great interest your manuscript, particularly as the dogma is that saturated fats are associated with CAD, making this a very interesting manuscript. Decanoic acid (capric acid) is a 10-carbon saturated fatty acid, however, you refer to it as a polyunsaturated fatty acid on line 51. Please correct as you refer to it later (discussion) as capric acid.
Several scientists have considered short and medium chain saturated fats as not causing CAD, with the longer chain saturated fats being the ones that contribute most to CAD. This manuscript then provides the first evidence (albeit observational) that certain saturated fatty acids may not be "bad". I would like to see you expand on this work.
Some minor comments:
Line 40: there are more than 2 keto type diets. Change sentence to reflect this.
Line 43: reference 8 is not formatted.
Line 45: "proven" is not used in this context, change to "shown".
Line 51: decanoic acid is saturated.
Line 87: the SI symbol for gram is "g" not "gm". SI symbols should be used in all scientific publications. Change throughout the manuscript.
Section 2.3: Specify that CAD is "self-reported". Have you considered statin use as an indicator of CAD?
Section 2.4: All diseases are "self-reported" not "physician's self-reported". Also confused here as you seem to combine self-reported and biochemical measures (which NHANES does have)? If you have FBG or A1c data, then why rely on self-reporting of diabetes for example? Please clarify what you used.
Author Response
- I read with great interest your manuscript, particularly as the dogma is that saturated fats are associated with CAD, making this a very interesting manuscript. Decanoic acid (capric acid) is a 10-carbon saturated fatty acid, however, you refer to it as a polyunsaturated fatty acid on line 51. Please correct as you refer to it later (discussion) as capric acid.
Reply: Thank you very much for reading my manuscript carefully and for your positive and valuable comments. I apologize for the confusion caused by incorrectly classifying decanoic acid as a polyunsaturated fatty acid. I have corrected it to “Decanoic acid, also known as caprylic acid, is a medium-chain saturated fatty acid commonly found in various dietary sources such as coconut oil, palm oil, dairy products, and fish (eel, salmon)” in the revised version of the manuscript. Thank you again for bringing this to my attention. (Line49-51)
- Several scientists have considered short and medium chain saturated fats as not causing CAD, with the longer chain saturated fats being the ones that contribute most to CAD. This manuscript then provides the first evidence (albeit observational) that certain saturated fatty acids may not be "bad". I would like to see you expand on this work.
Reply: As the reviewer notes, in the past, saturated fatty acids were widely recognized as one of the main causes of CAD because they can raise cholesterol levels causing atherosclerosis, however, recent studies have shown that there may be differences in the effects of different lengths of saturated fatty acids on the body. Although this study provides preliminary evidence, it is still an observational study and cannot establish a causal relationship. Therefore, we will conduct further experiments (animal and human levels) in the future to better understand the mechanism by which decanoic acid affects CAD.
- Line 40: there are more than 2 keto type diets. Change sentence to reflect this.
Reply: Thank you for pointing this out. The reviewer is correct, and we have changed “The ketogenic diet comes in two varieties” to “There are two common forms of ketogenic diets” in the discussion section of the revised manuscript.
- Line 43: reference 8 is not formatted.
Reply: We have revised.
- Line 45: "proven" is not used in this context, change to "shown".
Reply: Thank you very much for your attention to detail, we've changed “Research now proves that the ketogenic diet is …” to “As research has progressed, the ketogenic diet has been found to be …” as the reviewer commented. (Line 43)
- Line 51: decanoic acid is saturated.
Reply: We apologize for our mistake. We have corrected it to “Decanoic acid, also known as caprylic acid, is a medium-chain saturated fatty acid”. (Line49-50)
- Line 87: the SI symbol for gram is "g" not "gm". SI symbols should be used in all scientific publications. Change throughout the manuscript.
Reply: This suggestion is appreciated. We have carefully checked the manuscript and corrected the errors accordingly.
- Section 2.3: Specify that CAD is "self-reported". Have you considered statin use as an indicator of CAD?
Reply: Thank you for the helpful suggestions. As the reviewer said, statins can not only lower cholesterol levels but also prevent atherosclerotic plaque formation and anti-inflammatory effects, which are important in the treatment and prevention of CAD. However, although statins can increase the weight of a diagnosis of CAD, they do not play a decisive role. This is because statins can be used not only for simple hypercholesterolemia but also for the primary prevention of CAD and cerebral infarction. In addition, the NHANES database is not exhaustive in its description of drug use. Despite the limitations of using self-reported data, it is worth noting that questionnaires are an important part of national health and nutrition surveys, and many studies are based on questionnaire data.
- Section 2.4: All diseases are "self-reported" not "physician's self-reported". Also confused here as you seem to combine self-reported and biochemical measures (which NHANES does have)? If you have FBG or A1c data, then why rely on self-reporting of diabetes for example? Please clarify what you used.
Reply: We feel sorry for our carelessness. Our intent was to describe participants reporting that they had been diagnosed by a physician with a particular disease. In our resubmitted manuscript, we have corrected “as a physician's self-reported diagnosis” to “as a self-reported physician diagnosis”. Thanks for your correction. (Line 120-121)
Thank you very much for your question. We define disease based on relevant guideline recommendations or previous studies. For example, in the case of diabetes, we cannot determine whether a participant has diabetes based solely on FBG or HBA1c levels, as they may have already controlled both through the use of glucose-lowering medications and exercise. The same is true for hypertension.

Reviewer 3 Report
The introduction does not adequately present the significance/rationale of the research question or propose testable hypotheses based on proir literature. Please modify the introduction to include these components.
A significant majority of the introduction is dedicated to a discussion about ketogenic diets with little attention to its relevance to the research question. Greater attention to what is currently known about the role of DDA in CAD as well as the gaps (to be filled by this research) is needed. A brief reference to research about ketogenic diets, in support of the study findings (vs. a justification for conducting the research itself), is better suited for the discussion.
The reliability of the diet data is addressed as a limitation of the study; however, but this is only partially adequate. Please revise the methods to include a brief description of the validity of the measurement of DDA or similar with 2 (or less) 24-hour recalls.
It is unclear whether the DDA data were properly modeled to account for measurement error. Dietary intake is knowingly underreported. There are suggested modeling techniques for accounting for lack of reliability that do not appear to have been used here (e.g., energy adjusted intake). Please either specify how the diet data were handled to account this or re-run the analyses accordingly. (Major)
This was an epidemiological study of previously collected NHANES data. Please revise language suggesting that the authors "recruited" the participants throughout the manuscript, as needed. (Minor)
To my knowledge, not all NHANES participants provided biospecimens for laboratory analysis in NHANES (re: the inclusion of biomarker covariates). Please note this explicitly in the CONSORT table and table 1 (at least) and clarify how the covariates were handled if/when there were fewer participants with (covariate) biomarker data than diet/CAD data.
The authors neglect to address their definition of CAD as a study limitation. CAD was defined as having a self-reported history of MI, angina pectoris, or CAD diagnosis. Please address this in the discussion.
The conclusions should acknowledge the major study limitations and what steps would be necessary (or are being taken) to confirm the findings of this study.
The quality of English language was sufficient within the body of the manuscript. Some deficiencies were noted in the CONSORT table.
Author Response
- The introduction does not adequately present the significance/rationale of the research question or propose testable hypotheses based on proir literature. Please modify the introduction to include these components.
Reply: We are grateful to you for your effort in reviewing our paper and your valuable suggestions. Based on the reviewer's suggestions, we have reworked the introduction to the section: appropriately reduced the information on the ketogenic diet and appropriately increased the information on decanoic acid, such as its source, biological role, and research gaps on its association with CAD. The details are as follows:
The ketogenic diet is a high-fat, low-carbohydrate diet that was first used in the therapy of epilepsy. However, as research has progressed, the ketogenic diet has been found to be beneficial for a variety of diseases such as Alzheimer's disease, cancer, diabetes, and cardiovascular disease, but the major therapeutic mechanisms are not yet clear. In addition, several studies have shown a poor association between serum ketone levels and disease control. These results challenge the role of ketones themselves in disease control and raise the idea of a role for other components of the diet, particularly decanoic acid. Decanoic acid, also known as caprylic acid, is a medium-chain saturated fatty acid commonly found in various dietary sources such as coconut oil, palm oil, dairy products, and fish (eel, salmon). Decanoic acid is also one of the major components of the ketogenic diet and plays an important role in it. This is because it can be easily converted into ketones in the liver to serve as an alternative energy source for the brain and other tissues. Secondly, decanoic acid has unique properties compared to other fatty acids. Studies have found that decanoic acid has powerful anticonvulsant properties, which are especially beneficial for people with epilepsy. In addition, decanoic acid is thought to have improved mitochondrial function, antioxidant, and regulated blood lipid and glucose levels. Although these effects are also important in inducing coronary atherosclerosis, the specific effects of decanoic acid on CAD in individuals following a regular diet have not been thoroughly investigated. Therefore, this study aims to fill this research gap and provide insights into the potential role of decanoic acid in CAD. (Line 42-61)
- A significant majority of the introduction is dedicated to a discussion about ketogenic diets with little attention to its relevance to the research question. Greater attention to what is currently known about the role of DDA in CAD as well as the gaps (to be filled by this research) is needed. A brief reference to research about ketogenic diets, in support of the study findings (vs. a justification for conducting the research itself), is better suited for the discussion.
Reply: We thank the reviewer for pointing this out. As suggested by the reviewer, we have briefly described the ketogenic diet in the Introduction section (details are placed in the Discussion section) and added more information about decanoic acid. (Line 42-61)
- The reliability of the diet data is addressed as a limitation of the study; however, but this is only partially adequate. Please revise the methods to include a brief description of the validity of the measurement of DDA or similar with 2 (or less) 24-hour recalls.
Reply: Thank you very much for your helpful suggestions. We have added more details about DDA acquisition and calculation in the METHODS section: Participants were asked about all the food and beverages they had consumed in the last 24 hours and specific food scales and pictures were used to help participants estimate the amount of food they consumed. The obtained data will then be used to calculate each participant's energy, nutrient, and other food component intake through specialized software. (Line 87-91)
- It is unclear whether the DDA data were properly modeled to account for measurement error. Dietary intake is knowingly underreported. There are suggested modeling techniques for accounting for lack of reliability that do not appear to have been used here (e.g., energy adjusted intake). Please either specify how the diet data were handled to account this or re-run the analyses accordingly. (Major)
Reply: Thank you greatly for your very valuable advice. As noted by the reviewers, dietary intake has limitations of measurement error and the possibility of intentional underreporting. In order to minimize the impact they have on the results, we made the following efforts: First, we calculated the DDA mean by including data from two 24-diet recall interviews, thus reducing memory bias and measurement error. Second, as suggested by the reviewers, we included total energy, protein, fatty acids, fiber, and multivitamins as covariates in the regression analyses to more accurately assess the relationship between diet and coronary heart disease and to exclude the interference of other factors. The same is true of previous studies 1, 2. Although the ORs changed, the direction remained the same, suggesting that the findings were stable.
- This was an epidemiological study of previously collected NHANES data. Please revise language suggesting that the authors "recruited" the participants throughout the manuscript, as needed. (Minor)
Reply: Thank you for your reminder. Brought this issue to our attention and corrected it in the revised manuscript. We have corrected it to recruited (Line 78)
- To my knowledge, not all NHANES participants provided biospecimens for laboratory analysis in NHANES (re: the inclusion of biomarker covariates). Please note this explicitly in the CONSORT table and table 1 (at least) and clarify how the covariates were handled if/when there were fewer participants with (covariate) biomarker data than diet/CAD data.
Reply: Thanks for your meaningful comments. As the reviewers noted, not all NHANES participants provided complete data, and solutions for missing covariates were placed in the statistical analysis section: For missing categorical variables, we dealt with them by adding dummy variables (17.42% (n=6759) were missing for drinking status), and for missing continuity variables, we interpolated them using multiple interpolations. Overall, all data except for drinking status were only marginally missing (<10%). The distribution of the data after multiple interpolations compared to the raw data is shown in Table S1. As suggested by the reviewers, we have added this necessity statement to Table 1 as well: Missing treatment: categorical variables (adding dummy variables), continuous variables (multiple interpolations).
- The authors neglect to address their definition of CAD as a study limitation. CAD was defined as having a self-reported history of MI, angina pectoris, or CAD diagnosis. Please address this in the discussion.
Reply: Thanks for the heads up, we've added this statement to the discussion section: we used self-reported dietary intake data, which may be subject to memory bias and measurement error, and the same is true for the definition of CAD. (Line 291)
- The conclusions should acknowledge the major study limitations and what steps would be necessary (or are being taken) to confirm the findings of this study.
Reply: Your suggestion is very helpful to us. Following your suggestion, we have added more content to the conclusion section: However, because of the inherent limitations of cross-sectional studies and the specific limitations of the present study, well-designed prospective and experimental studies are needed in the future to validate the results of this study. (Line 301-303)
References
- Jun S, Cowan AE, Dodd KW, Tooze JA, Gahche JJ, Eicher-Miller HA, Guenther PM, Dwyer JT, Potischman N, Bhadra A, Forman MR, Bailey RL. Association of food insecurity with dietary intakes and nutritional biomarkers among US children, National Health and Nutrition Examination Survey (NHANES) 2011-2016. Am J Clin Nutr 2021;114:1059-69.
- Christensen K, Gleason CE, Mares JA. Dietary carotenoids and cognitive function among US adults, NHANES 2011-2014. Nutr Neurosci 2020;23:554-62.

Round 2
Reviewer 1 Report
The manuscript titled 'Study of the Relationship Between Dietary Decanoic Acid and Coronary Artery Disease: A Population-Based Cross-sectional Study' has undergone substantial improvements, with the authors addressing most of the feedback provided. However, I would like to share a few additional comments.
Abstract:
The authors noted that 'In stratified analyses, DDA was negatively associated with CAD prevalence in all populations except for the Non-Hispanic Black group.' However, according to Figure 3, a significant association between DDA and CAD is observed exclusively among the following subgroups: males, individuals of other Hispanic backgrounds, Non-Hispanic whites, those with an educational attainment of graduation or higher, individuals without hypertension, quit smokers, individuals who consume 10 or more alcoholic beverages per month, and those with a high GFR (glomerular filtration rate) of 60 or greater.
Methods:
1) It appears that breastfeeding women and participants with incomplete data on two 24-hour dietary recalls were not excluded from the analysis. Since their inclusion could potentially impact the study's findings, I recommend that the authors consider excluding these groups and conducting a separate analysis.
2) Physical activity has not been adjusted for or reported in Table 1. Given that physical activity is a crucial factor in assessing CAD, I suggest that the authors include adjustments for physical activity in their analysis and report the relevant information in Table 1.
3) It was not addressed whether participants with incomplete/unreliable diet records were included or not.
4) Multivitamin use has been added as a covariate, but physical activity was not. Is there reasoning for this?
Results:
1) In Table 2, please make the correction of 'gm/d' to 'g/d' to ensure consistency and accuracy in the units used."
Please thoroughly read through and check grammar.
Author Response
- The manuscript titled 'Study of the Relationship Between Dietary Decanoic Acid and Coronary Artery Disease: A Population-Based Cross-sectional Study' has undergone substantial improvements, with the authors addressing most of the feedback provided. However, I would like to share a few additional comments.
Reply: Thank you very much for your efforts to improve the quality of our manuscript and for recognizing our work. We have carefully studied your additional comments and have made the following necessary changes to the best of our ability.
- Abstract: The authors noted that 'In stratified analyses, DDA was negatively associated with CAD prevalence in all populations except for the Non-Hispanic Black group.' However, according to Figure 3, a significant association between DDA and CAD is observed exclusively among the following subgroups: males, individuals of other Hispanic backgrounds, Non-Hispanic whites, those with an educational attainment of graduation or higher, individuals without hypertension, quit smokers, individuals who consume 10 or more alcoholic beverages per month, and those with a high GFR (glomerular filtration rate) of 60 or greater.
Reply: Thank you for your careful review and helpful comments. We understand the reviewers' concerns about the results of our stratification analysis. Our study found that the DDA was negatively associated with CAD prevalence in all populations, but this does not mean that all subgroups were significantly associated. This may be due to differences in sample size, or individual hormone levels, lifestyle, and health status. We performed a reanalysis based on your follow-up suggestion after excluding participants with incomplete data on 24-hour dietary recalls. There were some new findings and the results of the stratified analysis have been rewritten according to your suggestions as follows: In subgroup analyses, the association between DDA and the prevalence of CAD remained significantly negative in the following subgroups: males, individuals of non-Hispanic whites, those with the educational attainment of graduation or higher, individuals with BMI of less than 25 kg/m2, individuals without hypertension, diabetes, quit smokers, individuals who consume 5 or more alcoholic beverages per month, and those with a high eGFR (estimated glomerular filtration rate) of 60 or greater. However, there was a significant interaction only for males and non-Hispanic whites (p-values for the interaction were 0.011 and 0.012, respectively). (Line 209-216). In the abstract, replace it with "Subgroup analyses found this relationship to be significant among males and non-Hispanic whites, and there was a significant interaction (interaction p-values of 0. 011 and 0. 012, respectively)". (Line 23-25)
- Methods:
3.1 It appears that breastfeeding women and participants with incomplete data on two 24-hour dietary recalls were not excluded from the analysis. Since their inclusion could potentially impact the study's findings, I recommend that the authors consider excluding these groups and conducting a separate analysis.
Reply: We deeply appreciate the reviewer’s suggestion. We agree with the reviewer that the inclusion of breastfeeding women and participants with incomplete 24-hour dietary recall data may have had an impact on the study results. However, as previously stated, complete breastfeeding data are not available in the NHANES database. This is similar to previously published studies and served as a limitation of the article. Therefore, we excluded only participants with incomplete 24-hour dietary recall data and conducted separate analyses.
In the new multivariate regression analysis, we still found a negative association between DDA and CAD prevalence, and these results are generally consistent with our original findings. Moreover, in the new subgroup analyses, it was found that in addition to race significantly modifying this relationship, so does gender. We have updated these results in the revised manuscript and explained them in the Discussion section.
Thanks again to the reviewers for their valuable suggestions, which were very helpful.
3.2 Physical activity has not been adjusted for or reported in Table 1. Given that physical activity is a crucial factor in assessing CAD, I suggest that the authors include adjustments for physical activity in their analysis and report the relevant information in Table 1.
Reply: Many thanks to the reviewer for your careful review and suggestions. As the reviewer notes, physical activity is indeed an important consideration when assessing the relationship between dietary nutrition and coronary heart disease. However, only NHANES survey cycles after 2010 contained data on physical activity. We reorganized and reanalyzed the data from the NHANES survey cycles that contained physical activity and found that the missing physical activity data remained significant. Out of a total sample of 16,903, 8,558 (50.63%) participants did not provide data on their physical activity.
We performed sensitivity analyses, dividing the data into two groups based on whether physical activity data were missing. All baseline characteristics were found to be significantly different between the two groups. As shown in the table below, participants in the no physical activity data group compared to the group with physical activity data were more likely to be female, older, less educated, lower income, lower eGFR, higher BMI, hypertension, diabetes, smokers, and drinkers. This means that the samples of the two groups have a large bias, which may have implications for the interpretation and generalization of the findings. We therefore did not conduct further analysis.
We also believe that this does not affect the results of the study. This is because the study also suggests that not all covariates associated with the dependent variable should be included in the multiple regression model. In addition, many previous studies examining coronary heart disease also did not include physical activity 1-3. Although physical activity may not be a necessary covariate for coronary heart disease research, our group will be aware of this in future studies. Thanks again for the advice!
Baseline characteristics of participants with and without physical activity data
|
Characteristics |
Data of physical activity |
P-value |
|
|
Unavailable |
Available |
|
|
|
N |
8558 |
8345 |
|
|
Age, years |
53.41 ± 17.09 |
46.83 ± 17.19 |
<0.001 |
|
Sex, % |
|
|
<0.001 |
|
Male |
3944 (46.09%) |
4199 (50.32%) |
|
|
Female |
4614 (53.91%) |
4146 (49.68%) |
|
|
Race, % |
|
|
<0.001 |
|
Mexican American |
1263 (14.76%) |
917 (10.99%) |
|
|
Other Hispanic |
950 (11.10%) |
745 (8.93%) |
|
|
Non-Hispanic White |
3280 (38.33%) |
3341 (40.04%) |
|
|
Non-Hispanic Black |
2033 (23.76%) |
1887 (22.61%) |
|
|
Other race |
1032 (12.06%) |
1455 (17.44%) |
|
|
Education level, % |
|
|
<0.001 |
|
<9th grade |
960 (11.22%) |
378 (4.53%) |
|
|
9–11th grade |
1395 (16.30%) |
626 (7.50%) |
|
|
High school |
2208 (25.80%) |
1576 (18.89%) |
|
|
College |
2605 (30.44%) |
2705 (32.41%) |
|
|
Graduate or above |
1390 (16.24%) |
3060 (36.67%) |
|
|
PIR |
2.20 ± 1.53 |
2.88 ± 1.66 |
<0.001 |
|
BMI, kg/m2 |
30.34 ± 7.56 |
28.72 ± 6.70 |
<0.001 |
|
Hypertension, % |
|
|
<0.001 |
|
No |
4151 (48.50%) |
5239 (62.78%) |
|
|
Yes |
4407 (51.50%) |
3106 (37.22%) |
|
|
Diabetes, % |
|
|
<0.001 |
|
No |
6503 (75.99%) |
7148 (85.66%) |
|
|
Yes |
2055 (24.01%) |
1197 (14.34%) |
|
|
Smoking Status, % |
|
|
<0.001 |
|
Never Smoking |
4492 (52.49%) |
5151 (61.73%) |
|
|
Quit Smoking |
1992 (23.28%) |
1806 (21.64%) |
|
|
Current Smoking |
2074 (24.23%) |
1388 (16.63%) |
|
|
Drinking status, % |
|
|
<0.001 |
|
Never Drinking |
1930 (22.55%) |
1497 (17.94%) |
|
|
1-5 drinks/month |
3093 (36.14%) |
3042 (36.45%) |
|
|
5-10 drinks/month |
364 (4.25%) |
593 (7.11%) |
|
|
10+ drinks/month |
670 (7.83%) |
985 (11.80%) |
|
|
Unknown |
2501 (29.22%) |
2228 (26.70%) |
|
|
eGFR, mL/min/1.73m2 |
91.08 ± 29.51 |
93.76 ± 24.81 |
<0.001 |
- Xu J, Eilat-Adar S, Loria C, Goldbourt U, Howard BV, Fabsitz RR, Zephier EM, Mattil C, Lee ET. Dietary fat intake and risk of coronary heart disease: the Strong Heart Study. Am J Clin Nutr 2006;84:894-902.
- Virtanen JK, Mursu J, Tuomainen TP, Voutilainen S. Dietary fatty acids and risk of coronary heart disease in men: the Kuopio Ischemic Heart Disease Risk Factor Study. Arterioscler Thromb Vasc Biol 2014;34:2679-87.
- Puaschitz NG, Strand E, Norekvål TM, Dierkes J, Dahl L, Svingen GF, Assmus J, Schartum-Hansen H, Øyen J, Pedersen EK, Drevon CA, Tell GS, et al. Dietary intake of saturated fat is not associated with risk of coronary events or mortality in patients with established coronary artery disease. J Nutr 2015;145:299-305.
3.3 It was not addressed whether participants with incomplete/unreliable diet records were included or not.
Reply: Thank you for the suggestion. We excluded participants with incomplete DDA data.
3.4 Multivitamin use has been added as a covariate, but physical activity was not. Is there reasoning for this?
Reply: Thanks for your attention to detail and comments! As mentioned earlier, we did not include physical activity as a covariate. The reasons are as follows: First, only NHANES survey cycles after 2010 contained data on physical activity. Second, A significant amount of data is still missing from the NHANES survey cycle including physical activity data. Out of a total sample of 16,903, 8,558 (50.63%) participants did not provide data on their physical activity. Third, we performed sensitivity analyses, dividing the data into two groups based on whether physical activity data were missing. All baseline characteristics were found to be significantly different between the two groups. In addition, many previous studies examining coronary heart disease also did not include physical activity.
- Results: In Table 2, please make the correction of 'gm/d' to 'g/d' to ensure consistency and accuracy in the units used."
Reply: Apologies for our mistake, we have corrected it.
- Please thoroughly read through and check grammar.
Reply: Thank you very much for your efforts to improve the quality of our manuscript. We have double-checked the full text and made corrections where necessary.

Reviewer 3 Report
Thank you for addressing a majority of the critiques I made. The remaining concern is in the introduction and there is an additional limitation that needs to be addressed.
Regarding the first, while the relevance of the ketogenic diet has been made clearer, the intro now lacks a relevant hypothesis and suggestion of potential scientific/public health impact that significant findings would have. Please consider adding a hypothesis statement and an impact statement that clearly communicates the need for the research.
Regarding the second, it needs to be noted that it cannot be concluded from the NHANES data whether a respondent was following a ketogenic diet; therefore, conclusions cannot be made about the impact of a ketogenic diet on CAD. It also remains unclear whether following a ketogenic diet is necessary to increase DDA to a level that would have beneficial effects on CAD.
Author Response
- Thank you for addressing a majority of the critiques I made. The remaining concern is in the introduction and there is an additional limitation that needs to be addressed.
Reply: Thank you very much for your efforts to improve the quality of our manuscript and for your positive evaluation of our work. We have carefully studied your additional comments and have made the following necessary changes.
- Regarding the first, while the relevance of the ketogenic diet has been made clearer, the intro now lacks a relevant hypothesis and suggestion of potential scientific/public health impact that significant findings would have. Please consider adding a hypothesis statement and an impact statement that clearly communicates the need for the research.
Reply: We deeply appreciate the reviewer’s suggestion. According to the reviewer’s comment, we have added more information about the research hypothesis and the significance of the study in the introduction section as follows: Given the trend toward younger age and the enormous public health burden of CAD, it is urgent that effective interventions are available for prevention. Combined with the beneficial effects of a KD and the ability of decanoic acid to influence a variety of CAD risk factors, we hypothesized that dietary decanoic acid (DDA) could reduce the incidence of CAD. If our hypothesis proves to be correct, it will provide a new dietary strategy for the prevention and treatment of CAD and thus have a significant impact on public health. (Line 56-62)
- Regarding the second, it needs to be noted that it cannot be concluded from the NHANES data whether a respondent was following a ketogenic diet; therefore, conclusions cannot be made about the impact of a ketogenic diet on CAD. It also remains unclear whether following a ketogenic diet is necessary to increase DDA to a level that would have beneficial effects on CAD.
Reply: We are grateful for the suggestion. As noted by the reviewer, this is important for readers to understand the significance and appropriate generalization of this study, and we have added these points to the Discussion section: Finally, it is important to note that because this study did not have data on whether participants were on a KD or not, it is not possible to conclude the effects of a KD on CAD. In addition, it is unclear whether a KD is needed to increase the DDA to levels that are beneficial for CAD. (Line 304-307)
